Osmotic response during kidney perfusion with cryoprotectant in isotonic or hypotonic vehicle solution

http://orcid.org/0000-0002-2650-937X Warner Ross M. 1
Yang Jun 1
Drake Andrew 1
Lee Youngjoo 1
Nemanic Sarah 2
Scott David 3
Higgins Adam Z. 1 adam.higgins@oregonstate.edu
1 School of Chemical, Biological, and Environmental Engineering, Oregon State University , Corvallis, Oregon , United States
2 Veterinary Radiology Consulting LLC , Lebanon, Oregon , United States
3 Department of Abdominal Transplantation, Oregon Health & Science University , Portland, Oregon , United States
Sotelo-Mundo Rogerio
Electronic publication date: 2023 Nov 21
Publication date: 2023
Volume: 11
Electronic Location ID: e16323
Received 2023 Aug 4; Accepted 2023 Sep 29
Copyright: © 2023 Warner et al.
Copyright year: 2023
Copyright holder: Warner et al.
License: This is an open access article distributed under the terms of the Creative Commons Attribution License, which permits unrestricted use, distribution, reproduction and adaptation in any medium and for any purpose provided that it is properly attributed. For attribution, the original author(s), title, publication source (PeerJ) and either DOI or URL of the article must be cited.
License URL: https://creativecommons.org/licenses/by/4.0/

Keywords: Cryoprotectant, Vitrification, Cryopreservation, Osmotic, Kidney, Organ, Vascular resistance, Computed tomography, Lactate dehydrogenase

Funding: A donation from the Cryonics Institute This work was supported by a donation from the Cryonics Institute. The funders had no role in study design, data collection and analysis, decision to publish, or preparation of the manuscript.

==============================
Organ cryopreservation would revolutionize transplantation by overcoming the shelf-life limitations of conventional organ storage. To prepare an organ for cryopreservation, it is first perfused with cryoprotectants (CPAs). These chemicals can enable vitrification during cooling, preventing ice damage. However, CPAs can also cause toxicity and osmotic damage. It is a major challenge to find the optimal balance between protecting the cells from ice and avoiding CPA-induced damage. In this study, we examined the organ perfusion process to shed light on phenomena relevant to cryopreservation protocol design, including changes in organ size and vascular resistance. In particular, we compared perfusion of kidneys (porcine and human) with CPA in either hypotonic or isotonic vehicle solution. Our results demonstrate that CPA perfusion causes kidney mass changes consistent with the shrink-swell response observed in cells. This response was observed when the kidneys were relatively fresh, but disappeared after prolonged warm and/or cold ischemia. Perfusion with CPA in a hypotonic vehicle solution led to a significant increase in vascular resistance, suggesting reduced capillary diameter due to cell swelling. This could be reversed by switching to perfusion with CPA in isotonic vehicle solution. Hypotonic vehicle solution did not cause notable osmotic damage, as evidenced by low levels of lactate dehydrogenase (LDH) in the effluent, and it did not have a statistically significant effect on the delivery of CPA into the kidney, as assessed by computed tomography (CT). Overall, our results show that CPA vehicle solution tonicity affects organ size and vascular resistance, which may have important implications for cryopreservation protocol design.

Introduction

Organ transplantation is one of the most important medical advances of the past century, offering lifesaving treatment to thousands of people each year. Despite a steady increase in the number of transplants performed, the number of people awaiting transplants continues to exceed the number of available organs (Schladt & Israni, 2021). Challenges with organ storage contribute to this problem (Giwa et al., 2017).

Currently, organs are stored at about 4 °C, which results in ischemic damage that worsens over time, limiting shelf life to a maximum of 4–6 h for heart and lung and up to 36 h for kidney. As a result, the logistics of organ transplantation must be managed within this short time window, including immunological matching and transportation from the donor to the recipient. If a transplant cannot be performed in time, the organ is discarded. About 20% of donated kidneys are discarded, in part because of concerns about ischemic damage during storage (Cooper et al., 2019).

Since the inception of organ transplantation in the 1950s, there has been interest in prolonging shelf life using cryopreservation. Despite significant advances over the last few decades, including reports demonstrating long term survival of cryopreserved rat and rabbit kidneys (Fahy et al., 2009; Han et al., 2023), large-scale organ cryopreservation remains a challenging problem. Recent efforts have focused on ice-free cryopreservation (i.e., vitrification), which requires the use of high cryoprotectant (CPA) concentrations and fast cooling and warming rates to suppress ice formation (Finger & Bischof, 2018). CPAs can be toxic, especially at high concentrations. Thus, a key challenge is the design of perfusion processes and solution compositions that enable delivery of high CPA concentrations without causing toxicity.

The problem of CPA toxicity is not unique to organs. To address this problem in isolated cells, we recently reported an approach to predict minimally toxic CPA equilibration methods using mathematical models of mass transfer, CPA toxicity, and osmotic damage (Benson, Kearsley & Higgins, 2012; Benson et al., 2018; Davidson, Benson & Higgins, 2014; Davidson et al., 2015). In the application of this approach to endothelial cell monolayers, we observed greater cell survival when we induced cell swelling during CPA addition using a hypotonic CPA vehicle solution, demonstrating the potential to leverage osmotically-induced cell size changes to reduce CPA toxicity (Davidson et al., 2015).

Osmotically-induced size changes have also been observed in organs, including kidney (Cady et al., 1966; Lachenbruch & Diller, 1999), heart (Butler et al., 2009; Kellen & Bassingthwaighte, 2003; Vargas & Johnson, 1967), lung (Wangensteen, Lysaker & Savaryn, 1977), brain (Fahy, Takahashi & Crane, 1984; McIntyre & Fahy, 2015), hindlimb (Rippe & Haraldsson, 1986; Wolf & Watson, 1989), and uterus (Farrant, 1964). For example, kidney perfusion with glycerol in isotonic vehicle solution has been reported to cause a decrease in kidney mass, whereas subsequent perfusion with isotonic solution caused mass to increase (Cady et al., 1966). Such size changes are qualitatively similar to those observed for isolated cells. As such, organ size changes may have implications for design of perfusion methods to reduce CPA toxicity.

The objective of this study was to examine the osmotic response of kidneys during perfusion with CPA in either isotonic or hypotonic vehicle solution. Our results provide insight into the physical processes occurring during CPA perfusion and will help inform design of perfusion processes to reduce toxicity during kidney cryopreservation.

Methods and Materials

Solution preparation

The base solution used in all experiments comes from Hauet et al. (2002) who proposed an isotonic extracellular-like hypothermic preservation solution containing polyethylene glycol (PEG) as an oncotic agent. We adopted this solution due to its simplicity and reported success with porcine kidneys (Faure et al., 2004; Hauet et al., 2002). The composition of this solution can be found in Table 1. All solutions were prepared within 24 h of experiments and were stored refrigerated in vessels with a small head space to minimize the potential for pH drift during storage.

Table 1 Composition of the isotonic extracellular-like base solution.

Constituent	Concentration	Vendor	
NaCl	118 mmol/L	EMD Millipore, Burlington, MA
VWR Chemicals BDH, Radnor, PA	
KCl	5 mmol/L	EMD Millipore, Burlington, MA	
NaHCO3	25 mmol/L	EMD Millipore, Burlington, MA	
MgCl2	1.2 mmol/L	VWR Chemicals BDH, Radnor, PA	
CaCl2	1.75 mmol/L	Fisher Chemical, Waltham, MA	
20 kDa PEG	30 g/L	Alfa Aesar, Haverhill, MA
Bean Town Chemical, Hudson, NH	
7.3 ± 0.1 pH	
330 ± 20 mOsm/kg	

The solution composition in Table 1 was denoted as the isotonic vehicle solution. Hypotonic vehicle solution was prepared using bicarbonate and PEG at the concentrations in Table 1, but without the other solutes. The osmolality of this solution was measured at ~50 mOsm/kg using an Advanced Micro Osmometer Model 3300 (Advanced Instruments, Norwood, MA, USA). CPA solutions were prepared by mixing ethylene glycol (Macron Fine Chemicals, Radnor, PA) or dimethyl sulfoxide (Me2SO; Fisher Chemical, Waltham, MA) with either isotonic or hypotonic vehicle solution.

Kidney acquisition

Porcine kidneys were obtained from two different slaughterhouses, Stanton’s Slaughter House (Albany, OR, USA) or Mohawk Valley Meats (Springfield, OR, USA). Kidneys from Stanton’s Slaughter House came from female Red Duroc pigs weighing 200–250 lbs. Kidneys from Mohawk Valley Meats came from female pigs of unknown breed weighing 250–350 lbs. For all porcine kidneys, there was an initial 10–15 min of warm ischemia due to the kidneys remaining in the animal during initial slaughterhouse processing. After resecting the kidney(s), the renal fascia was removed, ensuring that the renal capsule remained intact, and as much of the adipose capsule was removed as possible, except in the immediate vicinity of the renal hilum. Approximately two inches of the renal artery was kept with a specimen, as measured from the renal hilum, for perfusion setup purposes. The renal artery was then cannulated, and the kidney was cold-flushed with 500 mL of the isotonic extracellular-like solution of Table 1. Overall warm ischemia times ranged from 20–40 min, which is comparable to warm ischemia times reported in the literature (Ekser et al., 2018; Grosse-Siestrup et al., 2003; Unger et al., 2007). After the cold flush, the kidneys were placed in a cooler and brought to the laboratory. Subsequently, the kidneys were stored in a refrigerator at a temperature of 4 °C until they were used in an experiment. Two kidneys were subjected to long-term cold ischemia by storing them in the refrigerator for 5 days; for these kidneys, we augmented the flush solution with 200 IU/mL penicillin (Alfa Aesar, Haverhill, MA, USA). At the time of an experiment, kidneys were removed from the 4 °C environment and brought to room temperature.

Human kidneys were obtained from the Pacific Northwest Transplant Bank (Portland, OR, USA). The kidneys were procured from a brain-dead donor and were not used for transplantation because of unfavorable biopsy characteristics. The donor’s next of kin consented to the use of the kidneys for research. The donor was fully heparinized and the kidneys were flushed with University of Wisconsin solution and recovered with standard surgical techniques. The kidneys were stored at 4 °C for 72 h.

Perfusion apparatus

A gravity-fed perfusion apparatus was designed to maintain a hydrostatic pressure head of ~100 mmHg at the arterial inlet. This pressure is at the upper range of inlet arterial pressures in the literature (Fahy et al., 2004; Grosse-Siestrup et al., 2003; Post et al., 2012; Unger et al., 2007). For all experiments, there were at least two perfusates introduced to the kidney. Dedicated solution reservoirs were used for each perfusate, and the fluid level in each reservoir was maintained at a constant height by continuously pumping fluid to the reservoir and having a constant overflow. Each reservoir was connected to tubing that delivered fluid through a valve to one inlet of a Y joint located just upstream of the renal artery. The valves were used to switch between perfusates. When more than two perfusates were used, a series of Y joints were used to consolidate the perfusate lines into a single line, which could then be connected to the arterial cannula. To fully characterize the pressure in the perfusion system, we obtained a correlation for the pressure drop in the fluid network as a function of flowrate for each perfusate. This correlation was used to estimate the pressure at the renal artery, using the kidney influent flowrate calculated from the effluent flowrate and kidney mass change. Pressure at the renal artery ranged from about 60 to 100 mmHg.

Changes in kidney mass and effluent flowrate

To measure changes in kidney mass and effluent flowrate, a kidney platform was fabricated that rested on a scale (Model V11P3; Ohaus, Parsippany, NJ, USA). The kidney was placed on a grate on top of the platform, which allowed the effluent to drain into a funnel where it could be collected. Effluent was collected every minute and its mass was measured, giving an estimate of the effluent flowrate under the assumption that its density was equal to that of water. Kidneys were first equilibrated on a mass basis during perfusion with isotonic vehicle solution. Specifically, we looked for three consecutive kidney mass values during the equilibration period to be very similar before switching the perfusate to CPA solution. In initial experiments, the time required to achieve mass equilibrium was found to vary among kidneys from 12 to 27 min. As a result, for subsequent experiments, we used a 30 min equilibration period. During the equilibration period, the isotonic vehicle solution was recycled by returning the kidney effluent to the reservoir of the perfusion system. A total of 2 L of isotonic vehicle solution was used during the equilibration period. At the end of the equilibration period, the perfusate was switched to 10% m/v ethylene glycol in either the isotonic or hypotonic vehicle solution.

CT imaging of CPA transport during perfusion

Computed tomography (CT) imaging was used to assess changes in Me2SO concentration during kidney perfusion. Our approach is based on previous studies that have demonstrated a linear relationship between X-ray attenuation and Me2SO concentration (Corral et al., 2018a, 2015, 2018b; Han et al., 2020). An Aquilion 64 CT scanner (Toshiba, Tokyo, Japan) was used, with a voltage of 100 kV, a current of 200 mA, a scan time of 0.5 s, and a spatial resolution of 2 mm. All scans were exported as 8-bit grayscale images in jpg format.

We first confirmed a linear trend by testing varying concentrations of Me2SO in three different solutions: water, isotonic vehicle solution, and hypotonic vehicle solution. CT scans were acquired of 24-well plates containing these solutions, and the average grayscale value of the top cross-section of each well was quantified. As shown in Fig. S1, the average grayscale value increased linearly as the Me2SO concentration increased.

To assess changes in Me2SO concentration during kidney perfusion, two kidneys were placed in the CT scanner and imaged simultaneously. One kidney was perfused with 15% m/v Me2SO in isotonic vehicle solution, while the other was perfused with 15% m/v Me2SO in hypotonic vehicle solution. Three CT experiments were performed. In two of the experiments we used both kidneys from the same animal. In the third experiment, kidneys from different animals were used due to difficulties encountered during kidney acquisition at the slaughterhouse. CT scans were performed immediately before starting perfusion with Me2SO solution and at various time points afterwards.

For image analysis, we broke the kidney down into two spatial regions designated as the cortex and medulla. At every time point and for each kidney, we manually segmented the cortical and medullary regions. All image analysis was conducted using MATLAB, and we limited our analysis to only the coronal plane of the kidney—the simplest plane to determine the boundaries of the cortex, medulla, and pelvis. For one kidney, the alignment was such that we did not obtain a clear coronal plane. In this case, cortical and medullary regions were segmented using an oblique plane, and the pelvis was not segmented out as it was difficult to determine its border. Figure S3 is a representative image of this kidney, showing the boundaries of the cortical and medullary regions.

To estimate the Me2SO concentration in the cortical and medullary regions, the three middle CT slices were analyzed and the grayscale values within each region were averaged. The average grayscale values were used to estimate the Me2SO concentration as follows:

(1) ΔY=(1−xs)εMe2SOΔcMe2SO+(1−xs)Δ(vehiclesolutionbackground)

where ΔY is the change in grayscale value, xs is the solids volume fraction in the kidney, εMe2SO is the slope of the calibration curve (Fig. S1B), and ΔcMe2SO is the change in Me2SO concentration. The solids volume fraction of the kidney was assumed to be 30% based on previous studies (Blum et al., 2017; Bulger, 1987; Forbes, Cooper & Mitchell, 1953; Gardner & Vierling, 1969; Lee et al., 2018; Levitin et al., 1962; Reinoso, Telfer & Rowland, 1997). The effect of the assumed value of xs on the Me2SO concentration estimates is examined in Fig. S2. The final term on the right-hand side accounts for changes in the X-ray attenuation of the vehicle solution. For experiments with isotonic vehicle solution, no change is expected. However, the background for the hypotonic vehicle solution was about 15 gray levels lower than that of the isotonic vehicle solution (see Fig. S1A), which is equivalent to the gray level change for a 2.6% m/v change in Me2SO concentration. To account for this, we used the average of the isotonic and hypotonic vehicle solution backgrounds (i.e., 15/2 = 7.5) when computing the Me2SO concentration for kidneys perfused with hypotonic vehicle solution. This introduces an uncertainty of about ±1.3% m/v in estimates of Me2SO concentration.

Assessing osmotic damage

To estimate osmotic damage, we measured release of lactate dehydrogenase (LDH) into the effluent using an LDH assay kit (Catalog No. D2DH-100; BioAssay Systems, Hayward, CA, USA). All kidneys were first perfused with isotonic vehicle solution for 30 min. Kidneys subjected to hypotonic treatment were then perfused with hypotonic vehicle solution for 10 min, followed by perfusion with isotonic vehicle solution for 5 min. Control kidneys were perfused with isotonic vehicle solution for 10 min. The additional 5 min of isotonic perfusion after hypotonic treatment was included because of the observation that hypotonic treatment dramatically reduces flowrate. We hypothesized that some LDH might still be trapped in the kidney after the hypotonic perfusion period. Thus, we added 5 min of isotonic perfusion to increase the flowrate and allow us to capture any trapped LDH. As a final step, we introduced 10% (v/v) Triton X-100 (EMD Millipore, Burlington, MA, USA) in isotonic vehicle solution for 10 min, which is expected to kill all cells and release all remaining LDH into the effluent. Cell death due to hypotonic treatment was estimated by measuring the LDH released during the 10 min hypotonic perfusion and subsequent 5 min isotonic perfusion, and then dividing it by the total LDH released in the experiment. The corresponding cell death for control kidneys was estimated by dividing the LDH released during the 10 min isotonic perfusion by the total LDH released in the experiment.

Statistical analysis

Unless otherwise indicated, all data are reported using the mean and standard error of the mean. Different experimental groups were compared using t-tests with a significance level of 0.05.

Results

Kidney mass change

Figure 1 shows the change in kidney mass after initiating perfusion with 10% m/v ethylene glycol in either isotonic or hypotonic vehicle solution. Four different groups of kidneys were tested. Figure 1A shows the results for the kidneys with the shortest warm ischemia time, which are expected to be in the best condition. In this case, there is a clear trend showing a rapid decrease in mass after starting perfusion with CPA solution, followed by a more gradual mass increase. When hypotonic vehicle solution was used, the initial mass decrease did not last as long, and the subsequent mass increase was more substantial. Statistical analysis revealed significant differences between kidneys perfused with CPA in isotonic and hypotonic vehicle solution for all time points greater than 2 min.

Figure 1 Mass change after initiating perfusion with 10% m/v ethylene glycol in either isotonic (red squares) or hypotonic (blue circles) vehicle solution.

Four kidney groups were tested with different ischemia times, as indicated. Each curve represents mass measurements for a separate kidney. The gray shaded region denotes statistically significant differences based on t-tests comparing isotonic and hypotonic vehicle solutions at the different time points (p < 0.025). The average initial kidney mass was 197.4 ± 10.5 g, 219.8 ± 11.8 g, 173.3 ± 9.1 g, and 150.5 ± 5.8 g for A–D, respectively.

As shown in Figs. 1B–1D, the trends after initiating CPA perfusion become more muddled when ischemia time increases. Figure 1B shows results for kidneys with a slightly longer warm ischemia time. While some kidneys exhibited size changes similar to those in Fig. 1A, the trends were less consistent and some kidneys did not deviate much in mass throughout the perfusion. No statistically significant differences were detected between kidneys perfused using isotonic or hypotonic vehicle solution. Figures 1C and 1D shows results for porcine kidneys with 5 days of cold ischemia and human kidneys with 3 days of cold ischemia. Again, there are no obvious trends, and we were unable to detect any statistically significant differences between perfusion with isotonic or hypotonic vehicle solution.

Volumetric flowrate and vascular resistance

Figure 2 shows the effluent flowrate after initiating perfusion with 10% m/v ethylene glycerol in either isotonic or hypotonic vehicle solution. Results are shown for the same four groups of kidneys as in Fig.1, representing different levels of warm and cold ischemia. For kidneys perfused with CPA in isotonic vehicle solution, the flowrate trends varied depending on the level of ischemia. As shown in Figs. 2C and 2D, the two groups of kidneys with the longest cold ischemia exhibited an approximately 2-fold increase in flowrate after starting perfusion with CPA in isotonic vehicle solution, indicating a decrease in vascular resistance. However, most of the kidneys with shorter cold ischemia (Figs. 2A and 2B) did not exhibit such an increase in flowrate, and transient decreases in flowrate were even observed in some cases. In contrast, all four groups of kidneys perfused with CPA in hypotonic vehicle solution exhibited a significant and sustained decrease in flowrate, by as much as a factor of 10, indicating constriction of blood vessels and an increase in vascular resistance (Fig. S5). The flowrate trends for porcine kidneys with 5 days of cold ischemia (Fig. 2C) were very similar to the trends for human kidneys with 3 days of cold ischemia (Fig. 2D).

Figure 2 Change in effluent flowrate after initiating perfusion with 10% m/v ethylene glycol in either isotonic (red squares) or hypotonic (blue circles) vehicle solution.

Four kidney groups were tested with different ischemia times, as indicated. Each curve represents flowrate measurements for a separate kidney. The gray shaded regions denote statistically significant differences based on t-tests comparing isotonic and hypotonic vehicle solutions at the different time points. The p-values for the 3 min time points in panels (A and B) were 0.030 and 0.011, respectively. All other p-values were less than 0.01. The average flowrate at t = 0 was 82.3 ± 8.5 mL/min, 69.1 ± 13.1 mL/min, 10 ± 1.5 mL/min, and 18.8 ± 4.1 mL/min for A–D, respectively.

To test whether the increase in vascular resistance could be reversed, we perfused kidneys with 10% m/v ethylene glycol in hypotonic vehicle solution, then switched to 10% m/v ethylene glycol in isotonic vehicle solution. As shown in Fig. 3, the flowrate recovers within 5 min after switching to isotonic vehicle solution, indicating that the increase in vascular resistance caused by the hypotonic vehicle solution is reversible.

Figure 3 (A–B) The decrease in flowrate caused by hypotonic vehicle solution is reversible.

Kidneys were perfused with 10% m/v ethylene glycol in hypotonic vehicle solution for 10 min, then the perfusate was switched to 10% m/v ethylene glycol in isotonic vehicle solution. Two kidney groups were tested with different ischemia times, as indicated. Each curve represents flowrate measurements for a separate kidney. The data presented up until t = 10 min can be found in Fig. 2 as well.

Osmotic damage

Cell size changes induced by perfusion with CPA in isotonic or hypotonic vehicle solution could result in osmotic damage. To assess osmotic damage, we quantified LDH released into the effluent during perfusion with different solution compositions, relative to the total amount of LDH released after subsequent perfusion with surfactant to kill the cells. First, we examined the effects of perfusion with hypotonic solution (~50 mOsm/kg) in the absence of CPA. As shown in Fig. 4A, perfusion with hypotonic solution resulted in more cell death than perfusion with isotonic solution, but the difference was not statistically significant. Next, we examined perfusion of 10% m/v ethylene glycol in either isotonic or hypotonic (~50 mOsm/kg) vehicle solution. As shown in Fig. 4B, the resulting cell death measurements were comparable to those obtained in the absence of CPA, suggesting that hypotonic vehicle solution may be slightly damaging. However, once again the difference was not statistically significant. Overall, the amount of LDH released during hypotonic treatment was very low relative to the total LDH released after treatment with surfactant.

Figure 4 Estimated cell death due to osmotic damage during kidney perfusion.

(A) Kidneys were perfused with either isotonic or hypotonic solution in the absence of CPA (n = 3). (B) Kidneys were perfused with 10% m/v ethylene glycol in either isotonic or hypotonic vehicle solution (n = 4).

CPA distribution within the kidney

CT imaging was used to measure spatial and temporal changes in Me2SO concentration during kidney perfusion, using methods similar to those described in previous studies (Bleisinger et al., 2020; Corral et al., 2018a, 2015, 2018b). We perfused a total of six kidneys with 15% m/v Me2SO, three using isotonic vehicle solution and three using hypotonic vehicle solution. To analyze the resulting changes in Me2SO concentration, each kidney was separated into two different spatial regions, the cortex and the medulla, as illustrated in Fig. 5A. Overall, the results presented in Fig. 5 show that the Me2SO concentration increases with time, as expected. However, only the cortical region of one kidney reaches a steady-state at 15% m/v Me2SO, indicating incomplete equilibration with the perfusate for most of the kidneys. There is a large variation in the trends between different kidneys, especially for the kidneys perfused with CPA in isotonic vehicle solution.

Figure 5 Me2SO concentration estimated from CT images for porcine kidneys perfused with 15% m/v Me2SO in either isotonic (B) or hypotonic (C) vehicle solution.

(A) Shows a representative CT image. The Me2SO concentration was estimated separately in the cortex (solid symbols) and medulla (open symbols). Each kidney is represented by its own symbol, allowing comparison of medullary and cortical concentrations in the same kidney. For the hypotonic curves, there is an uncertainty of ±1.3% in the Me2SO concentration due to a difference in the background attenuation between the isotonic and hypotonic vehicle solution. The kidneys had a warm ischemia time of 25–40 min and a cold ischemia time of approximately 18 h.

The dramatic increase in vascular resistance observed during perfusion using hypotonic vehicle solution raises concerns about impaired CPA delivery into the kidney. To test this possibility, we statistically compared the measured Me2SO concentration at the 20 min time point for kidneys perfused using isotonic and hypotonic vehicle solution. This analysis did not reveal any significant differences (paired t-test, p = 0.90). However, the high variability in the data makes it difficult to rule out the possibility of impaired CPA delivery during perfusion using hypotonic vehicle solution. Inspection of the CT images did not reveal large dark regions, indicating that Me2SO appears to be reaching all regions to some extent, and the extreme scenario of large regions not being perfused does not seem to be the case.

Previous studies have reported challenges with delivery of CPA into the medulla (Fahy et al., 2004, 2009). Figures 5B and 5C shows that in most cases the Me2SO concentration in the medulla lags behind the cortex, which is consistent with these previous studies. A paired t-test yields a significant difference between the Me2SO concentration in the cortex and medulla for the 20 min time point (p = 0.024).

Discussion

Kidney mass change

Others have also observed a decrease in organ mass during CPA perfusion (Cady et al., 1966; Fahy, Takahashi & Crane, 1984; Lachenbruch & Diller, 1999; McIntyre & Fahy, 2015), and have attributed it to water movement down its osmotic gradient from the tissue into the capillaries, and out of the organ through the vasculature. The subsequent increase in mass is consistent with the relatively slow movement of CPA from the capillaries into the surrounding tissue. These organ mass trends are similar to the size changes expected for isolated cells, which have been observed to shrink after exposure to CPA due to loss of water, then regain volume as CPA and water enter the cells.

Our results suggest that these osmotically-driven organ size changes do not occur when capillary integrity is compromised as a result of ischemia. We did not observe consistent trends when warm ischemia was longer than 25 min, and the shrink-swell response completely disappeared when cold ischemia exceeded 3 days. We observed a significant increase in cell death after prolonged cold ischemia (Fig. S4), which is likely accompanied by a loss of capillary integrity. Previous studies provide support for the hypothesis that organ size changes are driven by movement of water and CPA across an intact capillary endothelium. Fahy, Takahashi & Crane (1984) observed a significant decrease in mass after perfusion of the rat brain with CPA (by as much as 50%), and attributed it to the hindered movement of CPA across the blood-brain barrier. McIntyre & Fahy (2015) later demonstrated that brain size changes could be eliminated by first perfusing with a surfactant to open the blood-brain barrier.

Volumetric flowrate and vascular resistance

Changes in vascular resistance have been widely observed in previous organ perfusion studies. Perfusion with CPA in isotonic vehicle solution has been observed to cause a modest decrease in vascular resistance in some cases (Pegg & Wusteman, 1977; Pegg et al., 1986), while other studies report negligible changes in vascular resistance (Jacobsen et al., 1978; Karow & Jeske, 1976). This is consistent with our results for perfusion with CPA in isotonic vehicle solution. To our knowledge, there are no previous studies of CPA perfusion in hypotonic vehicle solution. However, CPA washout has been reported to cause a striking increase in vascular resistance, by as much as 30-fold (Jacobsen et al., 1978; Pegg & Wusteman, 1977; Pegg et al., 1987). This is comparable to the increase in vascular resistance that we observed for perfusion with CPA in hypotonic vehicle solution.

To explain the observed changes in vascular resistance, previous studies have used a Krogh cylinder model to predict changes in capillary radius based on transport of water and CPA across the endothelium (Han et al., 2023; Pegg et al., 1986; Rubinsky & Cravalho, 1982). However, this model assumes the size of the Krogh cylinder is constant, which is equivalent to assuming the size of the organ is constant. As described above (see Fig. 1), organ size can change during CPA perfusion, which confounds interpretation of Krogh cylinder model predictions.

An alternative explanation for the changes in vascular resistance is cell volume changes induced by the CPA solution. In particular, CPA in hypotonic vehicle solution is expected to cause cell shrinkage, followed by swelling to an enlarged equilibrium cell volume. These swollen cells could impinge on the capillaries, decreasing their radius and increasing vascular resistance. This explanation is consistent with previous observations of a link between the increase in vascular resistance due to ischemia and histological observation of enlarged endothelial cells, as well as enlarged cells in the tissue surrounding the capillaries (Kloner, King & Harrington, 2018). If increased vascular resistance is caused by enlarged cells impinging blood vessels, then it should be possible to reverse this by perfusion with a solution that would cause the cells to shrink. In fact, perfusion with a hypertonic solution containing mannitol has been observed to mitigate the increase in vascular resistance caused by ischemia (Willerson et al., 1975). Our results are also consistent with this idea. Figure 3 demonstrates that switching to perfusion with CPA in isotonic vehicle solution restores vascular resistance to initial levels.

Comments on specimen quality and the slaughterhouse model

The slaughterhouse model is an attractive option as it allows many specimens to be acquired for a low cost and it does not require the sacrifice of additional animals. However, we have encountered some pitfalls with the slaughterhouse model that can lead to less accurate and potentially confounding results.

The first pitfall is that ischemia times can vary because of slaughterhouse processing constraints, which can lead to variability in organ quality and concomitant variability in experimental results. As shown in Fig. 6, ischemia time was found to be correlated with vascular resistance, suggesting that effluent flowrate can be used as a metric of kidney quality. Our results are similar to those of Unger et al. (2007), who reported a significant increase in vascular resistance as cold ischemia time increased from 0 to 2 to 24 h. Vascular resistance has also been observed to be higher in kidneys from slaughterhouse animals compared to kidneys from research animals (Grosse-Siestrup et al., 2003; Unger et al., 2007).

Figure 6 Effluent flowrate decreases as ischemia time increases.

Effluent flowrate was measured at the end of the isotonic perfusion equilibration period and normalized based on kidney mass.

Another pitfall is the lack of control over the animal at the time of slaughter. We were only able to obtain kidneys in a way that did not change the timing of the slaughterhouse’s processing schedule. In most cases, kidneys were resected by slaughterhouse personnel during the processing of the animal. This sometimes resulted in kidneys with only a short segment of the renal artery, which made cannulation difficult, and increased the risk of air bubbles entering the vasculature (Fig. S6). In addition, slaughterhouse processing is incompatible with pre-administration of anticoagulant, which is commonly done with research animals (Pegg & Wusteman, 1977; Pegg et al., 1986). This may have resulted in coagulation within the kidney. Even after flushing the kidneys with 500 mL of isotonic vehicle solution (which is on the higher end of other studies (Faure et al., 2004; Grosse-Siestrup et al., 2003; Hauet et al., 2002; Unger et al., 2007)), we still saw blood in the effluent during the subsequent 30 min perfusion equilibration period.

Implications for design of cryopreservation procedures

The observation that changes in organ size and cell size are analogous suggests that it may be possible to adapt cell cryopreservation strategies to organs. In particular, reduced toxicity is expected when CPA is delivered using hypotonic vehicle solution because it induces cell swelling and enables more CPA to be delivered for a given CPA concentration (Benson, Kearsley & Higgins, 2012; Davidson, Benson & Higgins, 2014; Davidson et al., 2015). Another strategy is to induce cell shrinkage after exposure to the final vitrification solution (Rall, 1987). This has two benefits. First, it permits shorter exposure to the vitrification solution, since cell shrinkage due to loss of water is faster than delivery of CPA across the cell membrane. Second, it makes it possible to quickly reduce the CPA concentration after rewarming by inducing cell swelling. Overall, the kidney size changes we observed in this study were small (±5%) compared to typical cell size changes, which suggests more modest potential for leveraging size changes to reduce CPA toxicity in kidneys. However, much larger size changes have been observed for the brain (Fahy, Takahashi & Crane, 1984). These results highlight the need for future studies to more thoroughly examine the implications of organ size changes for design of cryopreservation procedures.

Vascular resistance changes during CPA perfusion also have implications for design of organ cryopreservation procedures. It has long been recognized that increases in vascular resistance can occur during CPA removal, and that this can be mitigated using a hypertonic vehicle solution containing added mannitol (Pegg & Wusteman, 1977). To our knowledge, we are the first to examine perfusion with CPA in hypotonic vehicle solution. This resulted in a dramatic increase in vascular resistance that could be reversed by switching perfusion to CPA in isotonic vehicle solution. If the increase in vascular resistance is caused by cell swelling, then this indicates that the cells have equilibrated with the CPA. Exposure to CPA in hypotonic vehicle solution is expected to cause shrinkage, followed by swelling as CPA enters the cells. By the time the cells have swelled beyond their original volume, the intracellular CPA concentration is expected to be very close to the concentration of the external solution. It may be possible to take advantage of this phenomenon to customize the perfusion process during CPA addition, gradually increasing the CPA concentration each time the vascular resistance is observed to rise. This would allow the CPA concentration to be ramped up at a rate that aligns with the rate of cell equilibration with CPA.

Conclusions

In this work, we examined the effects of CPA vehicle solution tonicity on kidney size, vascular resistance, osmotic damage, and CPA delivery, providing insight into the physical processes occurring during CPA perfusion with implications for design of organ cryopreservation procedures. In particular, our results show that: Kidney perfusion with CPA causes a shrink-swell response analogous to that observed for cells. This is true for relatively fresh kidneys, but the trend disappears as warm and/or cold ischemia time increases, suggesting that the shrink-swell response only occurs when the capillary endothelium is intact.

Kidney perfusion with CPA in hypotonic vehicle solution causes vascular resistance to increase dramatically, which is consistent with reduced capillary diameter due to cell swelling. The increase in vascular resistance is reversible; switching to perfusion with CPA in isotonic vehicle solution returns the vascular resistance to initial levels.

Kidney perfusion with CPA in hypotonic vehicle solution does not cause significant osmotic damage as assessed from LDH released in the effluent.

Kidney perfusion with CPA in hypotonic vehicle solution does not cause major impairment of CPA delivery into the kidney, but it may cause more modest effects that we were unable to detect because of high variability in the data.

Overall, our results highlight the need for a better understanding of the physical processes occurring during CPA perfusion, including the effects of perfusate composition on changes in organ size, vascular resistance, osmotic damage, and delivery of CPA throughout the organ. Currently, a complete conceptual understanding of these interrelated factors is lacking, which makes it challenging to design organ perfusion procedures based on conceptual reasoning. Our results will help inform future efforts to improve design of CPA perfusion processes for organ cryopreservation.

Supplemental Information

Supplemental Information 1 Supplementary Material.

Click here for additional data file.

Supplemental Information 2 Raw Data.

Click here for additional data file.

We would like to thank both Stanton’s Slaughter House (Albany, OR, USA) and Mohawk Valleys Meats (Springfield, OR, USA) for allowing us to acquire porcine kidneys for our studies, as well as the Pacific Northwest Transplant Bank (Portland, OR, USA) for providing us with human kidneys. We would also like to thank Jason Wiest and Cynthia Viramontes for operating the CT scanner. Many thanks to Emi Ampo and Solomon Baez for laboratory assistance.

Additional Information and Declarations

Competing Interests

Author Contributions

Human Ethics

Data Availability

Sarah Nemanic is employed by Veterinary Radiology Consulting, LLC

Ross M Warner conceived and designed the experiments, performed the experiments, analyzed the data, prepared figures and/or tables, authored or reviewed drafts of the article, and approved the final draft.

Jun Yang performed the experiments, authored or reviewed drafts of the article, and approved the final draft.

Andrew Drake performed the experiments, authored or reviewed drafts of the article, and approved the final draft.

Youngjoo Lee performed the experiments, authored or reviewed drafts of the article, and approved the final draft.

Sarah Nemanic conceived and designed the experiments, analyzed the data, authored or reviewed drafts of the article, and approved the final draft.

David Scott conceived and designed the experiments, authored or reviewed drafts of the article, and approved the final draft.

Adam Z Higgins conceived and designed the experiments, performed the experiments, analyzed the data, prepared figures and/or tables, authored or reviewed drafts of the article, and approved the final draft.

The following information was supplied relating to ethical approvals (i.e., approving body and any reference numbers):

Human kidneys were obtained from a deceased donor. The donor’s next of kin provided consent to use the organs for research.

The following information was supplied regarding data availability:

The numerical data is available in the Supplemental Files.

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
