# Peer review of "Osmotic response during kidney perfusion with cryoprotectant in isotonic or hypotonic vehicle solution"

_PeerJ, doi:10.7717/peerj.16323_

## Round 0.1 · original submission · Minor Revisions

Please submit a revised version soon, considering the reviewers' promising and thorough comments.

Reviewer 1 ·

Basic reporting

Clearly written. Literature generally well cited but additional references suggested in details below. Figures are acceptable.

Experimental design

Meets the aims and scope of the journal

Validity of the findings

Findings are generally valid with minor qualifications discussed below.

Additional comments

Osmotic response during kidney perfusion with cryoprotectant in isotonic or hypotonic vehicle solution
General comments:
This work introduced CPA perfusion with hypotonic carrier solution and compared it with traditional CPA perfusion with isotonic carrier solution, with respect to osmotic response. The idea was adapted from the cell/tissue work to reduce the chemical and mechanical (osmotic) toxicity, and this paper presented that using hypotonic carrier solution didn’t cause significant osmotic damage nor significantly impair CPA delivery. The latter point was not made clear which requires either more experimental data support or explanation. Further extension of this work to non-penetrating agents as described in multiple publications from Fahy and others would have also been informative, but perhaps these can be at least considered in the discussion. While there are some limitations, this work will be informative in the rapidly developing field of organ and tissue cryopreservation.
Major comments:
1. In Figure 5, the concentration of CPA using isotonic (Fig. 5B) and hypotonic (Fig. 5C) looked very different at the end of the perfusion, even considering the uncertainty of DMSO concentration due to the attenuation difference. Moreover, the kidneys in Fig. 5B seemed all equilibrated because the curves all stabilized, however the kidneys in Fig. 5C seemed still increasing at the end of the perfusion, which may be an explanation for the CPA difference. In summary, I don’t think Figure 5 can support the conclusion that “Kidney perfusion with CPA in hypotonic vehicle solution does not significantly impair CPA delivery into the kidney”. More experimental data or more discussion are needed in the manuscript. Some suggestions can be found in minor comment 3.
Minor comments:
1. Line 62, when mentioning recent advance, I suggest adding the citation of the recent rat kidney vitrification report (Han, Nature Comm, 2023), and modify the statement into “large-scale organ cryopreservation remains a challenging problem”. When perfusing large-scale organs, since it requires even higher concentrations of CPA, therefore optimizing perfusion protocol is very critical, which is also even more highlighting the importance of this work, exploring using hypotonic carrier solution to optimize the CPA perfusion.
2. Line 166/Figure S1, using grayscale values directly is acceptable in general, however, the CT machine might have some fluctuation from scan to scan. Another popular approach in CT study, especially in CPA related research, is Hounsfield Unit (HU), where during each scan, a water and air sample are scanned together with the sample, and those two signals were used as the calibration points. HU approach can increase the reproducibility of the CT scans, which can be considered to use in the follow up studies.
3. Line 304, calibration curves presented in Figure S1 are for CPA in solutions, not in tissues. Please refer to this paper, Han, Advanced healthcare materials, 2020, where in Figure 2, one can tell that the solution and equilibrated tissues follow different linear trends, and tissues have lower signals, which may explain your results. The kidneys may be actually equilibrated with the DMSO after 60 min perfusion, but you fit it to the solution curve shown in Figure S1, instead of the tissue calibration curve, that’s why you are getting false lower concentrations. This may also can explain the difference between Fig 5 B and C. So, I suggest doing a series of CT scans for the tissues equilibrated with different concentrations of DMSO and obtain the calibration curve for tissue, then fit the perfusion CT data to this obtained tissue curve, and compare the concentrations.
4. Line 357, I suggest adding the reference of Han, et al. "Model-Guided Design and Optimization of CPA Perfusion Protocols for Whole Organ Cryopreservation." Annals of Biomedical Engineering (2023): 1-13, since it is the most recent work that used Krogh cylinder model. In that work, they didn’t consider the volume change, as you correctly pointed out, however they did observe shrink-swell behavior in flow resistance. This current work together with that work can both provide insights into building the next generation model to more accurately predict the physical behavior during CPA perfusion.

Reviewer 2 ·

Basic reporting

The authors' writing is excellent. There were few if any grammatical or typographical issues. This reviewer would prefer to see equation 1 typeset so that the subscript Me2SO was not italics.

The references were sufficient and appropriate.

The structure of the manuscript was appropriate, the background sufficient for this reviewer, and the data were provided in an easy to assess format.

The results were clear and self contained.

Experimental design

The article presents original primary research within the Aims and Scope of the journal, focused on both exploratory and mechanistic study of cryoprotectant equilibration in whole kidneys.

The research question is well motivated by the prior research, both from addressing key gaps in the literature from previous studies in kidneys, but also from the authors’ prior work in addressing optimal CPA equilibration in cells, monolayers and tissues.

The study is very well designed with careful attention to detail, and practical acknowledgement of experimental challenges, including the use of abattoir obtained tissues. The inclusion of some human kidneys is particularly admirable and provides a useful dataset for implementation of other work using porcine models for human kidney perfusion.

Item for addressing: The authors could do more to address any differences or similarities in their results between the porcine and human models. This paper will be better utilized if there are concrete conclusions about the relevance of the key metrics explored in this work to comparative model species to human organ perfusion.

Validity of the findings

The findings are in line with hypotheses and informative in the scope of the broader literature. This is clearly stated.

The data have been provided, and are easily independently analyzed.

Item for addressing: The authors state they use a t-test to compare between treatments. This seems not likely to be appropriate for time-series data especially since there are multiple other factors, including ischemia time and species. Were t-tests used for all time points? Were t-tests used for comparisons among ischemia time or species? It seems that in either case, there is a danger of multiple comparisons, but either way there should be much more clear explanation for how treatment responses were compared.

Item for addressing: Along these lines the gray regions in figures 1 and 2 is not clear. Is this on a point by point basis, or for grouped time points?

Item for addressing: The authors create standard curves for CT scans in a 24 well dish in both vector media. Is this sufficient to expect that the response will be the same in tissues? It would be ideal to have fully equilibrated kidney tissue at, say 0, 5, 10, 15% m/v to ensure that concentrations align with standard curves. Perhaps the authors can justify this with their existing dataset?

Additional comments

This paper by Warner et al presents a rational approach towards optimizing cryoprotectant equilibration in in-tact kidneys. The authors have a track record of optimization in individual cell and small tissue piece CPA equilibration optimization, and this is a natural well executed adaptation into whole organs. Their previous work showing that hyposmotic medium is an ideal strategy for minimal toxic loading of CPA provides the background for a number of hypotheses explored here. First, they carefully explore the relationship between CPA equilibration and kidney weight and add interesting information about the importance of cold ischemia time, with reasonable hypotheses about damaged vasculature as a cause for the lack of classic “shrink-swell” type behaviour. Next they explore the relationship between loading dynamics and carrier medium osmolality, demonstrating that there is a clear effect of carrier medium on both the weight dynamics and flow rate manifested as vascular resistance and that this resistance is reversible, suggesting that hyposmotic medium swells surrounding tissues constricting vessels. Next they show that there is no relationship between vehicle solution and cell death, suggesting that were hypotonic medium reduce toxicity, it would not be associated with additional cell death due to osmotic effects. Finally, they perform technically challenging experiments using CT to identify the completeness of CPA equilibration as a function of time. In this, they demonstrate that there are no statistical differences between vector media. This is a bit surprising given the significant difference in flow rates, but the authors do admit that their data have high variability.

---

## Round 0.2 · accepted · Accept

Thanks for addressing the revisions requested. Now, your manuscript is accepted in PeerJ.